# Patatin-Related Phospholipase *AtpPLAIIIα* Affects Lignification of Xylem in Arabidopsis and Hybrid Poplars

**DOI:** 10.3390/plants9040451

**Published:** 2020-04-03

**Authors:** Jin Hoon Jang, Ok Ran Lee

**Affiliations:** Department of Applied Plant Science, College of Agriculture and Life Science, Chonnam National University, Gwangju 61186, Korea; jinhun92@naver.com

**Keywords:** Arabidopsis, poplar, phospholipase, lignin, phloroglucinol, xylem

## Abstract

Lipid acyl hydrolase are a diverse group of enzymes that hydrolyze the ester or amide bonds of fatty acid in plant lipids. Patatin-related phospholipase AIIIs (pPLAIIIs) are one of major lipid acyl hydrolases that are less closely related to potato tuber patatins and are plant-specific. Recently, overexpression of ginseng-derived *PgpPLAIIIβ* was reported to be involved in the reduced level of lignin content in *Arabidopsis* and the mature xylem layer of poplar. The presence of lignin-polysaccharides renders cell walls recalcitrant for pulping and biofuel production. The tissue-specific regulation of lignin biosynthesis, without altering all xylem in plants, can be utilized usefully by keeping mechanical strength and resistance to various environmental stimuli. To identify another pPLAIII homolog from *Arabidopsis*, constitutively overexpressed *AtpPLAIIIα* was characterized for xylem lignification in two well-studied model plants, *Arabidopsis* and poplar. The characterization of gene function in annual and perennial plants with respect to lignin biosynthesis revealed the functional redundancy of less lignification via downregulation of lignin biosynthesis-related genes.

## 1. Introduction

Plant phospholipase A (PLA) families are classified into two groups: low-molecular-weight PLA_2_s and patatin-related PLAs (pPLAs), which hydrolyze an acyl ester bond of glycerolipids to liberate free fatty acids and lysophospholipids [1,2]. By hydrolyzing membrane glycerolipids, pPLAs participate in many aspects of cellular biology such as signal transduction, cell growth regulation, membrane remodeling in response to environmental stresses, and lipid metabolism [2,3]. The *Arabidopsis AtpPLAs* consist of 10 member of genes and are classified into three subclasses based on the gene structure and amino acid sequence similarity: *AtpPLAI*, *AtpPLAIIα, β, γ, δ,* and *AtpPLAIIIα, β, γ, δ* [2,4]. *AtpPLAI* and *AtpPLAIIs* have been reported to be involved in the plant response to biotic stresses, auxin signaling, and phosphate deficiency. *pPLAIII* overexpression (OE) showed a similar stunted and dwarf mutant phenotype, and resulted in an altered cell wall content of cellulose [5,6,7] or lignin [8,9]. 

The molecular and biochemical functions of patatin-related phospholipases such as *AtpPLAIIIβ* and *AtpPLAIIIδ* were characterized and were found to cause short and stunted phenotypic characteristics with increasing phospholipids and galactolipids in *Arabidopsis* [5,6,10]. The altered lipid species were reported to result in the reduction of cellulose content [5,6]. However, another close homolog *AtpPLAIIIα* was only functionally characterized in rice and showed a similar phenotype to *AtpPLAIIIβ-OE* and *AtpPLAIIIδ-OE*, but the contents of many lipid species were lower than that of wild-type (WT) species with decreased cellulose content [7]. To clarify these conflicting lipidomics results, we have recently characterized another *pPLAIII* homolog from ginseng especially focused on cell wall composition. Overexpression of *PgpPLAIIIβ* still resulted in a stunted and dwarf phenotype similar to that of *AtpPLAIIIβ-OE* and *AtpPLAIIIδ-OE* [5,10]. However, we additionally demonstrated that the ginseng-derived *PgpPLAIIIβ* was involved in the reduction of lignin content when overexpressed in *Arabidopsis* and poplar [8,9]. The *PgpPLAIIIβ* displays 77% amino acid sequence identity with *PtpPLAIIIβ* from *Populus*, and 61.5% with *AtpPLAIIIβ* and 61.3% with *AtpPLAIIIα* from *Arabidopsis* [8]. It is not clear whether the *PgpPLAIIIβ* is a functional homolog of *AtpPLAIIIα* or *AtpPLAIIIβ* in *Arabidopsis*. Thus, the function of native *AtpPLAIIIα* in cell wall composition in endogenous *Arabidopsis* still remains to be determined.

In this study, a functional characterization of *AtpPLAIIIα* in *Arabidopsis* and perennial poplar, focused on the composition of cell walls, is carried out by generating overexpression lines. Phloroglucinol lignin staining and direct lignin quantification demonstrated that *AtpPLAIIIα* might be involved in the alteration of lignin content and composition in two model plants. 

## 2. Results

### 2.1. Isolation of Knockout Mutant and Overexpression Lines of *AtpPLAIIIα*

Xylem tissue staining using phloroglucinol-HCl by overexpressing ginseng-derived *PgpPLAIIIβ* showed much reduced lignin [8,9] which suggested the possible involvement of other close homologs. Considering *AtpPLAIIIα* is the closest homolog (Figure 1A) among *Arabidopsis*, ginseng and poplar, where the genetic transformation system is well-established [8,9], a functional characterization of the *AtpPLAIIIα* gene was carried out. SALK T-DNA insertion mutant line (SAIL830G12) [11] was obtained and further confirmed that it is a knockout (KO) mutant (Figure 1B,C). Constitutive overexpression lines of *AtpPLAIIIα* under the 35S cauliflower mosaic virus promoter with an mRFP (monomeric red fluorescent protein)-tagged end of C-terminal region were generated by *Agrobacterium*-mediated floral dipping method [12]. Among several lines following mendelian segregation, three *OE* lines were further chosen for further lignification analysis (Figure 1B).

### 2.2. Overexpression of *AtpPLAIIIα* Reduced Lignin Content in Arabidopsis

Lignin consists of three building blocks, the three monolignols namely *p*-coumarylalcohol, coniferyl alcohol, and sinapyl alcohol [13]. Phlologlucinol staining provides clues to the extent of cinnamaldehydes in the plant tissues and thus, indicates the general distribution of cinnamaldehydes differentiating other aromatics [14]. 

All lines were stained red with phloroglucinol-HCl, but the staining was weaker in *AtpPLAIIIα-OE* lines; the weakest staining was found in the highly expressing *AtpPLAIIIα-OE* #13 line (Figure 2A). This was consistent with the results obtained through direct quantification, showing that lignin was not altered in the *AtpplaIIIα KO* mutant but decreased significantly in strong *OE* lines in leaf and stem (Figure 2B). The expression of two transcription factors involved in lignin biosynthesis [15], *AtMYB58* and *AtMYB63*, was also concomitantly reduced (Figure 2C,D). This suggests that *AtpPLAIIIα* functions upstream of two transcriptional activators. Lysolipid species or fatty acids released by AtpPLAIIIα may also regulate the function of downstream *AtMYB58* and *AtMYB63*. Previously, it was reported that the cellulose content was increased in a knock-out mutant of *AtpPLAIIIβ,* and decreased in *OE* lines, which showed a small and stunted phenotype [5], similar to that of *AtpPLAIIIα-OE*. However, only lignin content, and not cellulose (Figure 2E), was decreased in lines with increased *AtpPLAIIIα* expression.

### 2.3. Lignin Biosynthesis Genes are Decreased in *AtpPLAIIIα-OE*, and It is Related to Ethylene Biosynthesis

To further confirm the reduced lignin content in *AtpPLAIIIα-OE* (Figure 2B), several genes involved in lignin biosynthesis [16] have been investigated. In agreement with the reduced lignin staining, lignin biosynthesis-related genes (Figure 3A) were all downregulated in *OE* lines, and did not changed significantly in *KO* mutant lines (Figure 3B). Coniferaldehyde 5-hydroxylase *(F5H*), a cytochrome P450-dependent monooxygenase, is known to play a key role in the formation of S (syringyl) units of lignin. Since two *OE* lines showed an increased expression of the *F5H-OE* line, the differential regulation of S-lignin can be expected.

Reduced and thick hypocotyl observed in *pPLAIII-OE* seedlings [5,6,8,9] resembles the phenotype by alternation of ethylene biosynthesis. Thus, several key genes involved in ethylene biosynthesis were additionally analyzed in *KO* mutant and several *AtpPLAIIIα-OE* lines. The ethylene biosynthesis pathway is simply established by the production of *S*-adenosyl-methionine (*S*-AdoMet) and 1-aminocyclopropane-1-carboxylic acid (ACC) as the precursors. The first committed step in ethylene biosynthesis is the conversion of *S*-AdoMet to ACC by ACC synthase (*ACS*), and finally ACC is oxidized by ACC oxidase (*ACO*) to form ethylene [17] (Figure 3C). *AtACS11* gene was upregulated in *KO* mutant and downregulated in all *OE* lines (Figure 3D). Whereas, two *AtACO2* and *AtACO4* genes were increased in both mutant and *OE* lines (Figure 3D), which indicates that the modulation of *AtpPLAIIIα* expression affects the ethylene biosynthesis. 

### 2.4. Overexpression of *AtpPLAIIIα* in Poplar Reduced the Plant Height and Lignification in Xylem

As a model for perennial woody plants, the poplar tree offers another possibility to address questions that cannot be easily answered in *Arabidopsis* [18]. In plant, *Arabidopsis* has been used as the prime model organism and an impressive number of genetic studies were performed using available tools and techniques. *Arabidopsis* is more closely related to *Populus* than other dicot taxa [18] and possesses advantageous traits for comparative genetic study. However, *Arabidopsis* is an unusual plant in many aspects. Especially since the lignification process that can be characterized in secondary xylem is limited. Thus, we have generated transgenic poplars by *Agrobacterium*-mediated gene transformation [19]. The full-length genomic DNA sequence of *Arabidopsis AtpPLAIIIα* was heterologously expressed in hybrid poplar under the control of the 35S promoter. The efficiency of transformation in poplar showed a regeneration rate of 10% (Appendix A). Insertion of *AtpPLAIIIα* gene was confirmed by genomic DNA Polymerase Chain Reaction (PCR) using vector specific and gene specific primers (Appendix A). Two strongly expressing *OE* lines (#4 and #6) were selected for further analysis (Figure 4A). All transgenic poplars displayed reduced plant height with smaller and darker green leaves (Figure 4B–E), which is the reminiscence phenotype of *PgpPLAIIIβ-OE* in *Arabidopsis* and poplar [8,9]. Plant cell walls are composed of a complex matrix of three organic compounds: cellulose, hemicellulose, and lignin. When poplar stems grown for 24 weeks were analyzed for direct quantification of carbohydrate profiles, total lignin was decreased in two *OE* lines (Figure 5A). The reduced lignin was mostly from acid insoluble rather than soluble (Figure 5A). Other contents of cell wall constituents such as glucan (cellulose) and xylan (hemicellulose) were not changed (Figure 5B).

### 2.5. MYB Transcription Factors and Lignin Biosynthesis Genes are All Downregulated in *AtpPLAIIIα-OE* of Poplar 

The coordinated activation of the secondary cell wall biosynthesis is known to be regulated by NAC domain transcription factors and MYB-mediated transcriptional networks [20]. The NAC domain transcription factors bind directly to the promoters of several wood-associated transcription factors, such as MYBs [21]. Two MYB transcription factors, *At*MYB58 and *At*MYB63, are well-known specific transcriptional activators of lignin biosynthesis during secondary cell wall formation in *Arabidopsis* [9,15,22]. Among several MYB transcription factors, *PtrMYB152* and *PtoMYB92* were reported to be involved in lignin biosynthesis in poplar [8,23,24]. The transcriptional levels of *PtrMYB92* and *PtoMYB152* were downregulated 43% and 53% compared to WT independently (Figure 6A). Relevant monolignol biosynthetic genes including phenylalanine ammonia-lyase 1 (*PAL1*), cinnamate-4-Hydroxylase 1 (*C4H1*), 4-coumarate-CoA ligase (*4CL*)*,* Hydroxycinnamoyltransferase 1 (*HCT1*)*,* 4-coumarate 3-hydroxylase 3 (*C3H3*)*,* caffeoyl-CoA O-methyltransferase 1 (*CCOAOMT1*)*,* cinnamoyl-CoA reductases 2 (*CCR2*)*,* catechol-O-methyltransferase 2 (*COMT2*)*,* and cinnamyl alcohol dehydrogenase 1 (*CAD1*) were all significantly downregulated in xylem of all transgenic plants overexpressing *AtpPLAIIIα* (Figure 6B). It suggests that *AtpPLAIIIα-OE* is involved in the negative regulation of lignification in poplar.

### 2.6. Lignin Autofluorescence and Expression of YUCCA8 and ERF Genes in *AtpPLAIIIα-OE*

Plant tissues contain natural fluorophores in cell walls, which is called autofluorescence and can be used for tissue imaging [25]. The most practical approach in terms of using autofluorescence as a histological tool is to use excitation with a range of wavelengths over the ultraviolet (UV) and visible light spectrum. Lignin is characteristic of xylem tissue containing a variety of fluorophore [26]. Thus, autofluorescence by confocal laser scanning microscopy has been utilized for lignin imaging [25]. Autofluorescence of xylem tissue was analyzed based on three sections including outer xylem where young tissues are present and middle and inner xylem that are mostly mature tissues [27]. UV autofluorescence was significantly decreased in middle xylem of *OE* lines (Figure 7A), whereas no difference was observed in outer and inner xylem (Appendix A).

Flavin monooxygenases (YUCCA) are one of the key enzymes involved in indole-3-acetic (IAA) biosynthesis, and two of 11 members, *YUCCA8* and *YUCCA9*, reveal a positive link between auxin signaling and lignification when overexpressed [8,28]. In the poplar genome database, only *YUCCA8* homolog was identified [8], and its transcripts were decreased in basal xylem of stem in *AtpPLAIIIα-OE* (Figure 7B). Overexpression of *YUCCA8* and *YUCCA9* led to strong lignification of aerial plant tissues, and it is linked to increased levels of ethylene production [28]. Constitutive ectopic expression of *AtpPLAIIIα* resulted in downregulation of lignin biosynthesis-related gene expressions (Figure 3B), and modulated mRNA levels related to ethylene biosynthesis (Figure 3D). To go one step further, ethylene-responsive marker genes such as *ERF1* and *AtERF14* [29] in *Arabidopsis* as well as *PtEFR1* and *PtERF14* in poplar were quantified by qRT-PCR in different organs and were all confirmed to be upregulated by *AtpPLAIIIα* overexpression (Figure 8). It suggests that at least in stem of *AtpPLAIIIα-OE,* the lignification is related with the ethylene biosynthesis.

## 3. Discussion

Native *AtpPLAIIIβ* and *AtpPLAIIIδ* from *Arabidopsis* [5,6] and *OspPLAIIIα* from rice [7] are all shown to have a reduced amount of cellulose without knowing the exact mechanism when overexpressed in endogenous plant systems. However, close to another homolog from ginseng, *PgpPLAIIIβ,* rather displayed reduced content of lignin by phloroglucinol staining and direct lignin quantification analysis by overexpressing in *Arabidopsis* and poplar [8,9]. In this study, endogenous *AtpPLAIIIα-OE* also displayed a reduced amount of lignin instead of cellulose content alteration (Figure 2), which is antagonistic to that found in rice focused on cell wall composition [7]. The reduction of lignin content was further confirmed in the perennial model plant poplar (Figure 5). There seemed to be no direct correlation of lipid species that caused the different cell wall composition since *AtpPLAIIIβ-OE* and *AtpPLAIIIδ-OE* showed increased lipid and fatty acid species [5,6] and *OspPLAIIIα* responded in reverse ways [7]. However, the alternation of lipid species and/or fatty acids could possibly affect the central carbon flux involved in cell wall biogenesis, which definitely needs to be elucidated in the near future. 

Plant hormone auxin is fundamental for the regulation of various aspects of plant growth and development, including cell elongation, division, differentiation, tropism, senescence, abscission, apical dominance, and flowering [1,30]. Patatin-related phospholipase has been shown to be involved in the response of plants to auxin signaling [2]. Our previous study also showed that *PgpPLAIIIβ* is involved in the transcriptional regulation of several auxin response-related genes such as *Aux/IAA*, *GH3*, *SMALL UP RNA* (*SAUR*), and *YUCCA* genes by constitutive overexpression of *PgpPLAIIIβ* [9]. YUCCA encodes a flavin monooxygenase (FMO)-like enzyme which catalyzes hydroxylation of the amino group of tryptamine in tryptophan-dependent auxin biosynthesis [31]. Among 11 members of *YUCCA* gene family, two of them, *YUCCA8* and *YUCCA9* were reported to be involved in lignification of plant aerial tissues when overexpressed [28]. The transcripts of *PtYUCCA8* were reduced by 24% in the basal xylem part of the poplar stem by overexpression of *AtpPLAIIIα* (Figure 7B). Recently we also reported that *PgpPLAIIIβ-OE* also downregulated *PtYUCCA8* expression by 25% in xylem and resulted in decreased lignin content [8]. This *YUCCA*-mediated lignification is supposed to be linked to increased levels of ethylene production [28]. By *AtpPLAIIIα-OE*, we observed upregulation of genes located in the last step of ethylene-biosynthesis (Figure 3B). Ethylene responsive marker genes such as *ERF1* and *AtERF14* [29] were all upregulated in seedling, leaves, and 7-week-old stems of *Arabidopsis* (Figure 8A). The transcripts of homologous genes from poplar, *PtERF1* and *PtEFR14,* were also increased (Figure 8B). This suggests that the reduced lignin content could also be affected by the modulation of ethylene biosynthesis not directly via the *YUCCA8* dependent auxin signaling pathway. Of course, direct measurement of ethylene will shed more light on further clear understanding. The cross talks between *YUCCA*-mediated auxin signaling and ethylene biosynthesis seems to be differentially regulated in *AtpPLAIIIα-OE.* The modulation patterns of gibberellin biosynthesis genes by the expression levels of *PgpPLAIIIβ* [8] is another criterion to be considered to understand these mechanisms and how they function in cell wall composition.

The primary plant growth can occur from two distinct populations of stem cells, called shoot apical meristem (SAM) and root apical meristem (RAM). All aerial shoot and underground root structures are derived from these two meristems [32]. Final formation of these new organs is coordinated tightly with procambium formation and the specification of vascular tissue. Especially in woody plants such as poplar where a secondary growth pattern becomes prominent, extensive secondary xylem (wood) is formed [32]. The procambium provides a source of vascular stem cells during primary growth, and the vascular cambium cells perform an analogous role during the secondary growth of every year. The procambial cell specification and organization are thought to be regulated by the canalization of auxin fluxes [33]. Lignin autofluorescence from xylem tissue of *AtpPLAIIIα-OE* in poplar was shown to be restricted to the outermost part of cell walls and had a much weaker signal compared to that of WT (Figure 7A). The transcripts of *PtYUCCA8* that are involved in auxin biosynthesis were also downregulated an average of 24% in the stem of poplar *AtpPLAIIIα-OE* (Figure 7B). While auxin is considered an essential factor for vascular tissue specification, it is also necessary to consider that other plant hormones are also playing important roles during vascular tissue development [34]. These results suggest that *AtpPLAIIIα* regulates the lignification of mature xylem rather than developing xylem [27] in poplar via regulating plant hormone metabolism. 

Lignin is a complex aromatic biopolymer of monolignols and is polymerized at the surface of the cell walls [16]. Lignin provides mechanical reinforcement for a plant to stand upright and enables long-distance water transport [35]. The genes involved in the phenylpropanoid pathway of flavonoid biosynthesis and lipid biosynthesis showed clear functional enrichment, and are supported by the acetate pathway [36]. This suggests that the functional activity of phospholipase involved in lipid metabolism could possibly affect the phenylpropanoid-based polymer-like lignin biosynthesis.

## 4. Materials and Methods

### 4.1. Plant Materials and Growth Conditions

The *Arabidopsis thaliana* ecotype Columbia (Col-0) and hybrid poplars (*Populus alba* × *Populus glandulosa*) were used as model wild-type (WT) plants in this study. The SALK T-DNA insertion *AtpplaIIIα* mutant (SAIL830G12) was obtained from the *Arabidopsis* stock center. Seeds sown on half-strength MS medium (Duchefa Biochemie, The Netherlands) containing 1% sucrose and 0.8% agarose were transplanted to an autoclaved soil mixture containing soil, vermiculite, and pearlite (3:2:1 v/v/v) under long-day conditions of 16 h light/8 h dark at 23 °C. Hybrid poplars were maintained and propagated on MS medium [37] containing 3% sucrose and 0.2 mg L^−1^ indole-3-butyric acid. The in vitro propagated poplars were acclimated in the soil, and further grown at controlled growth conditions at 24 °C under long day conditions. 

### 4.2. Transgenic Construct and in planta Transformation 

The full-length DNA sequence of *AtpPLAIIIα* from *Arabidopsis* was cloned into the modified pCAMBIA1390 vector [1] under the control of the cauliflower mosaic virus *35S* promoter. The transgenic construct was confirmed by nucleotide sequencing and transformed into *Arabidopsis* and hybrid poplar using *Agrobacterium tumefaciens* C58C1 (pMP90) [38]. Simplified floral dipping method [12] was used for transformation in *Arabidopsis*, and transformants were selected on hygromycin-containing plates (50 µg/mL). In planta transformation in poplar (Appendix A) was carried out by co-cultivating the stem segments of the hybrid poplar (*Populus alba* × *Populus glandulosa*) with *Agrobaterium tumefaciens* (C58C1) carrying the construct, following the previous report [8,19]. 

### 4.3. Total RNA Isolation and PCR Amplification

Total RNA was extracted using the Pure Link^TM^ RNA Mini Kit (Invitrogen, California, CA, USA), according to the manufacturer’s instructions [8,9]. The quantification of total RNA was performed by a Nano-MD UV-Vis spectrophotometer (Scinco, Seoul, Korea). Complementary DNA (cDNA) was synthesized using RevertAid Reverse transcriptase (Thermo Scientific, Waltham, MA, USA) in a 20 μL reaction volume. Quantitative reverse transcription (qRT) polymerase chain reaction (PCR) was performed and analyzed using TB Green™ Premix Ex Taq™ (Takara, Shiga, Japan) and Thermal Cycle Dice real-time PCR system (Takara, Shiga, Japan) as reported previously [9]. Three independent replicates were performed for each primer sets. The primer sequences used in this study are provided in Appendix A.

### 4.4. Acetyl Bromide Soluble Lignin Assay for Total Lignin Quantification 

The acetyl bromide method for quantification of lignin was conducted following the previous report [39]. Seven-week-old stem and leaf tissues were pooled, ground in liquid nitrogen, freeze-dried for 48 h, and filtered through a 425 μm screen. Each 10 mg of dried materials were washed with 95% EtOH four times and distilled water twice to remove soluble components. After 12-h drying at 60 °C, products were dissolved in 2 mL of acetyl bromide:glacial acetic acid (1:3, v/v) and incubated at 70 °C for 30 min. Thereafter, 0.9 mL 2 M NaOH, 3 mL glacial acetic acid, and 0.1 mL 7.5 hydroxylamine-HCl were added sequentially. Supernatant was obtained by centrifuging at 4000 g for 10 min. The supernatant was diluted 20-fold with glacial acetic acid and measured at 280 nm using a spectrophotometer (Scinco, Seoul, Korea).

### 4.5. Analysis of Carbohydrate Profiles

Whole trimmed stems of 24-week-grown poplar were air-dried and ground in a mill to obtain 20–80 mesh wood particles. Wood particles (20–80 mesh) were degreased via extraction with benzene:ethanol (2:1, v/v) for 6 h at 80 ℃. The resulting wood particles were used for total lignin content determination [40] that were quantified by the individual sum of acid-insoluble (Klason lignin) and acid-soluble lignin. Each sample weight (0.3 g) was hydrolyzed with 3 mL of 72% (w/w) H_2_SO_4_ at 30 ℃ for 1 h, filled up with distilled water to reach final weight 89 g (final 4% of H_2_SO_4_), and autoclaved at 121 ℃ for 1 h. The final particles were then filtered through a glass filter and dried overnight at 105 ℃. Acid-insoluble lignin was directly weighted by measuring the excess on the filter. Acid soluble lignin and carbohydrate were quantified from the filtrates. To analyze the content of acid soluble lignin, filtrates were fill up with water to reach 100 mL and quantified using spectrophotometer at 205 nm (Shimadzu UV-2401 PC UV-VIS). For the quantification of carbohydrate, the filtrates were adjusted at pH 5.5 ~ 6.0 using CaCo_3_, and filtered via 0.45 μm sized syringe filter to remove impurities. Then the contents of glucan, xylan, and mannan were estimated by HPLC (Waters 2695 System; Alliance, MA, USA). The HPLC system was operated with an Aminex HPX-87P column (300 × 7.8 mm, Bio-Rad, Hercules, CA, USA) and refractive index detector (Waters 2414 system; Alliance, MA, USA).

### 4.6. Detection of Lignin Autofluorescence

Using a rotary microtome (RM2235, Leica, Germany), the stem of 24-week-old poplar was sectioned in a transverse axis to a thickness of 70 μm. Autofluorescence of the lignin was then observed by confocal laser scanning microscopy (TCS SP5 AOBS/Tandem, Leica Germany) under ultraviolet (UV) light. The images were acquired at the Korea Basic Science Institute, Gwangju. Korea.

## Figures and Tables

**Figure 1 plants-09-00451-f001:**
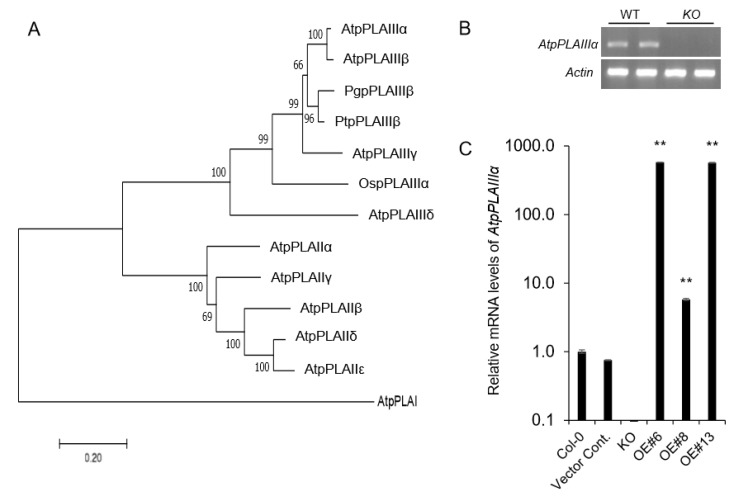
Phylogenetic analysis, T-DNA insertion knockout mutant, and overexpression lines of *AtpPLAIIIα* gene. (**A**) Phylogenetic tree of AtpPLAs, OspPLAIIIα, PgpPLAIIIβ, and PtpPLAIIIs proteins. The phylogenetic tree was constructed using the Mega7 program (7.0.26 version, neighbor-joining method). *At, Arabidopsis thaliana; Os, Oryza sativa; Pg, Panax ginseng; Pt, Populus trichocarpa*. (**B**) RT-PCR of *AtpPLAIIIα* gene expression in wild-type (WT) and *AtpplaIIIα* knockout (*KO*) lines. *β-Actin* was used as internal control. (**C**) Transcript levels of *AtpPLAIIIα* genes in *KO* and overexpression (*OE*) lines as measured by qRT-PCR. Data represent the mean ± SE of three independent replicates at *P* < 0.05 (*) and *P* < 0.01 (**), respectively.

**Figure 2 plants-09-00451-f002:**
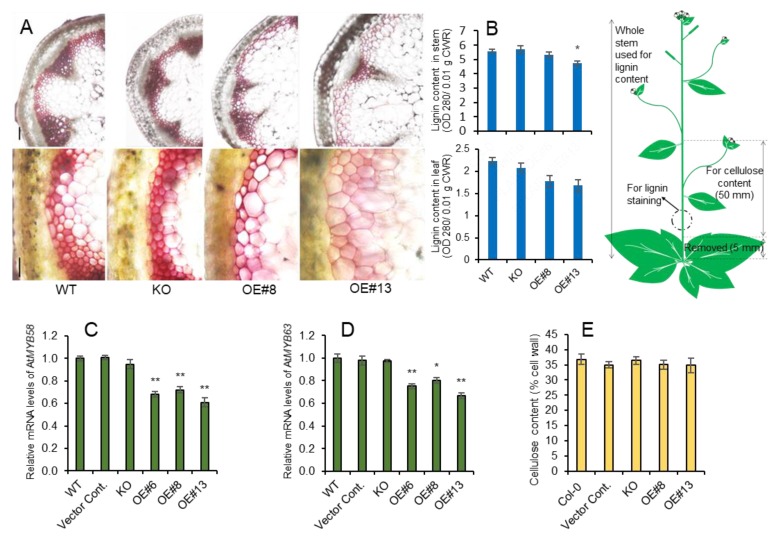
Reduced lignin content in stems of *AtpPLAIIIα-OE* did not affect cellulose content. (**A**) Histochemical staining of cross sections from 7-week-old stems from WT, *KO*, and *OE* lines using Phloroglucinol-HCl. Scale bars = 100 µm. (**B**) Lignin content of 7-week-old stem and leaf assessed by acetylbromide. (**C**,**D**) Relative gene expression of transcription factors, *AtMYB58* and *AtMYB63*, involved in lignin biosynthesis was decreased. The Ct value for each gene was normalized to that for *β-actin* and was calculated relative to a calibrator using the formula 2^-ΔΔCt^. (**E**) Cellulose content of 7-week-old stems of WT, *KO*, and *OE* lines was not changed; n = 5. Data (**B**–**D**) represent the mean ± SE of three independent replicates at *P <* 0.05 (*) and *P <* 0.01 (**), respectively.

**Figure 3 plants-09-00451-f003:**
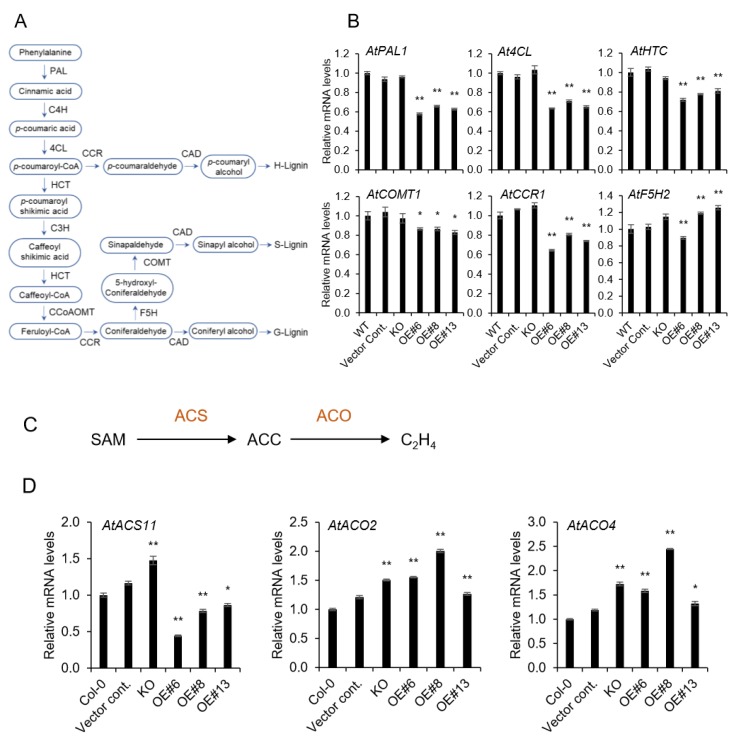
Expression levels of lignin and ethylene biosynthetic genes by qRT-PCR. (**A**) Lignin biosynthesis pathway, and (**B**) relative mRNA levels of genes related to lignin biosynthesis; *PAL*: phenylalanine ammonia-lyase 1; *C4H*: cinnamate-4-hydroxylase; *4CL*: 4-coumarate-CoA ligase; *HCT*: hydroxycinnamoyltransferase; *C3H*: *4-*coumarate 3-hydroxylase; *CCoAOMT*: caffeoyl-CoA O-methyltransferase; *CCR*: cinnamoyl-CoA reductases; *COMT*: catechol-O-methyltransferase; *F5H*: coniferaldehyde 5-hydroxylase; *CAD*: cinnamyl alcohol dehydrogenase. (**C**) Simple schematic diagram of ethylene biosynthesis pathway and (**D**) mRNA levels of genes related to ethylene biosynthesis; *SAM*: S′adenosyl methionine, *ACC*: 1-aminocyclopropane-1-carboxylic acid. Data represent the mean ± SE of three independent replicates at *P <* 0.05 (*) and *P <* 0.01 (**), respectively.

**Figure 4 plants-09-00451-f004:**
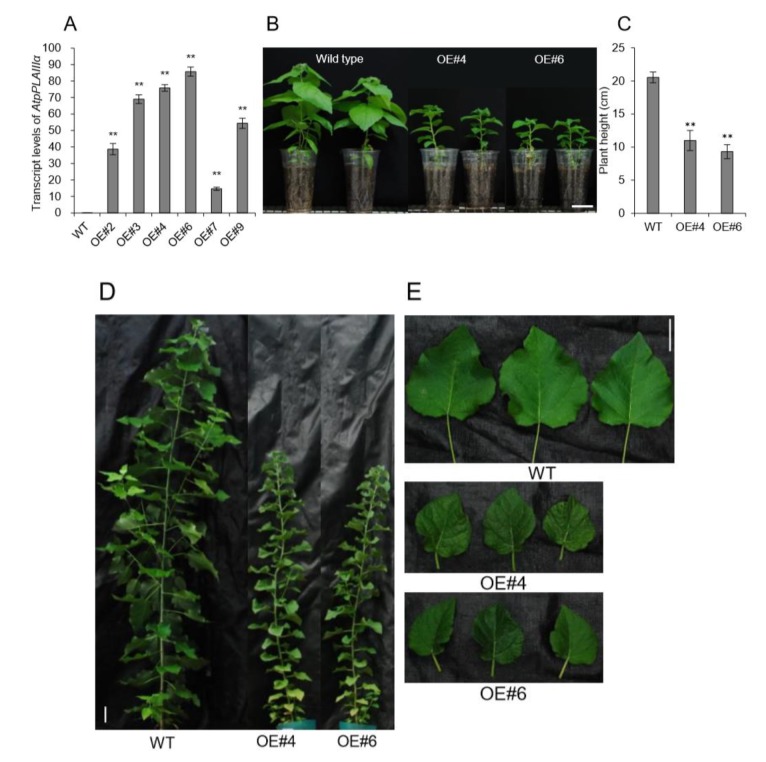
Overexpression of *AtpPLAIIIα* in poplar resulted in a dwarf phenotype. (**A**) Transcript level of *AtpPLAIIIα* in WT and *AtpPLAIIIα-OE* poplars; n = 3. (**B**) Growth phenotype and (**C**) plant height of 3-month-grown poplars (WT, n = 10; OE, n = 5). (**D** and **E**) Six-month-grown poplar tree phenotypes in WT and *OE* lines. Bars = 5 cm. Data represent the mean ± SE of three independent replicates at *P <* 0.01 (**).

**Figure 5 plants-09-00451-f005:**
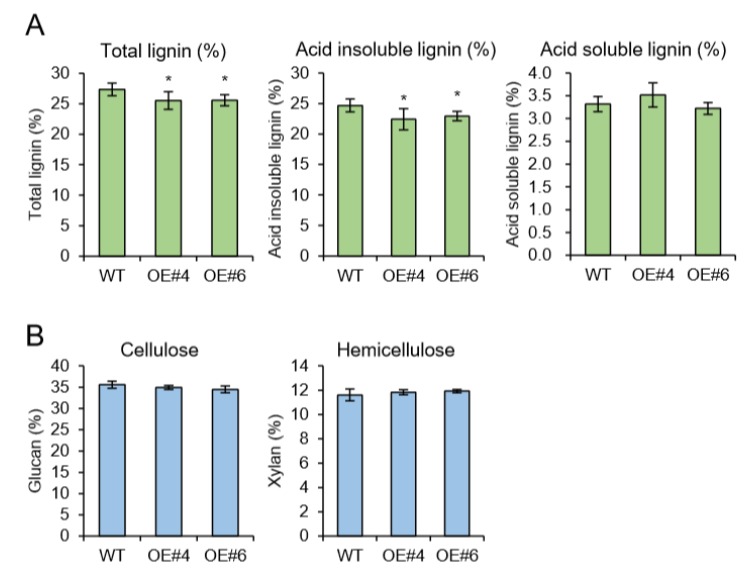
Cell wall carbohydrate ratio analysis (Klason assay) in 6-month-grown poplar. (**A**) Total lignin (acid insoluble) was decreased in OE lines. (**B**) Glucan and xylan ratios were not changed by overexpression of *AtpPLAIIIα*. Data represent the average ± SE from multiple independent replicates; n = 9 (WT), n = 3 (OE) at *P <* 0.05 (*).

**Figure 6 plants-09-00451-f006:**
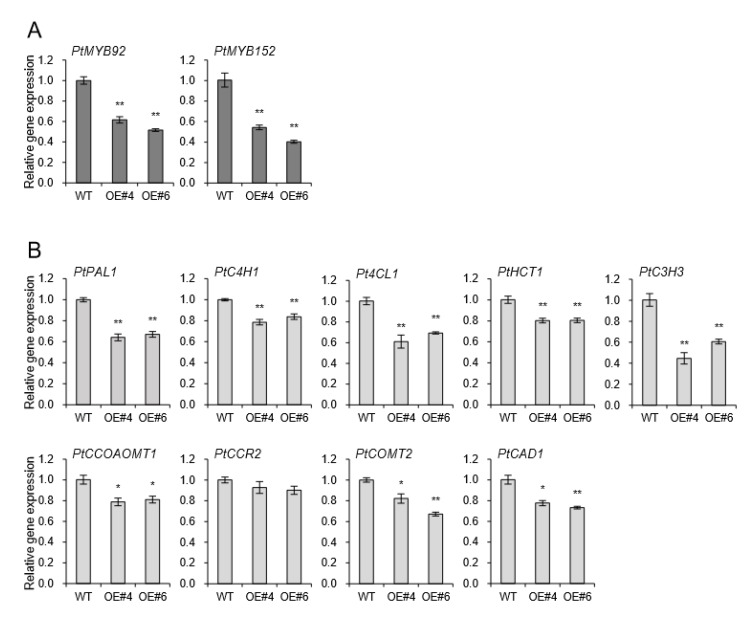
Genes involved in lignin biosynthesis were down-regulated in xylem of *AtpPLAIIIα-*OE lines. Expression levels of (**A**) two transcriptional activators, *PtMYB92* and *PtMYB152,* and (**B**) several lignin biosynthetic genes. Each datum represents the average ± SE from three independent replicates at *P <* 0.05 (*) and *P <* 0.01 (**), respectively.

**Figure 7 plants-09-00451-f007:**
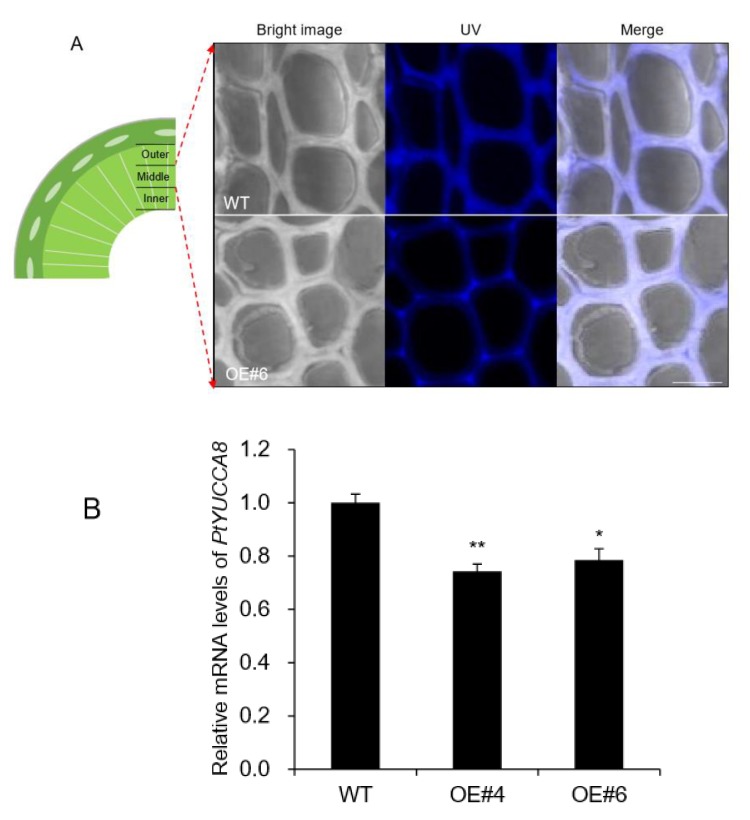
The middle xylem layer was altered in *AtpPLAIIIα-*OE. (**A**) Enlarged images of portions of the middle xylem showed that the cell walls were thickened in OE, whereas the UV autofluorescence signal was weakened compared to that of WT. Bright image, UV, and merged images of WT and OE were displayed. Bar = 10 μm. (**B**) Expression levels of *PtYUCCA8* in basal xylem part of stem in WT and *AtpPLAIIIα*-OE. Each datum represents the average ± SE from three independent replicates at *P* < 0.05 (*) and *P* < 0.01 (**), respectively.

**Figure 8 plants-09-00451-f008:**
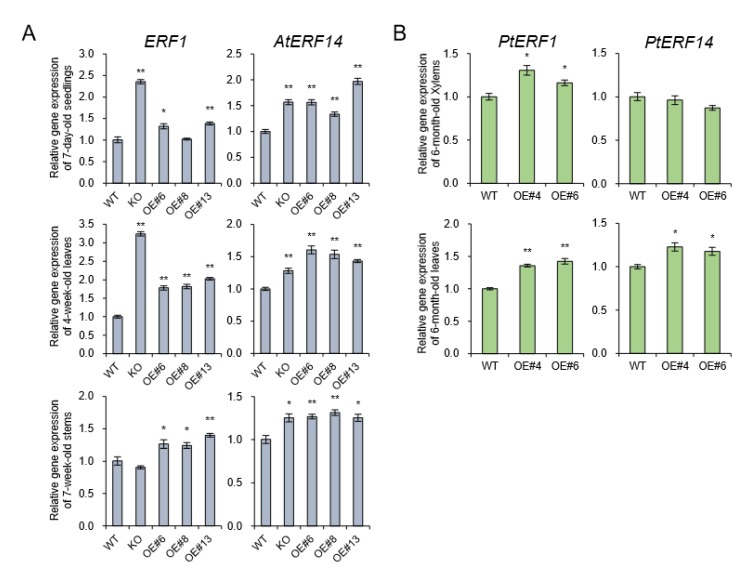
Expression of ethylene-responsive genes in *Arabidopsis* and poplar. (**A**) The transcript levels of *ERF1* and *AtERF14* which is upregulated by ethylene [29] were quantified by qRT-PCR in seedlings, leaves, and stems of *AtpPLAIIIα*-OE lines of *Arabidopsis*. (**B**) Expression levels of *PtERF1* and *PtERF14* in stems of *AtpPLAIIIα*-OE lines of poplar. Each datum represents the average ± SE from three independent replicates at *P* < 0.05 (*) and *P* < 0.01 (**), respectively.

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
