# Peer review of "Patatin-Related Phospholipase AtpPLAIIIα Affects Lignification of Xylem in Arabidopsis and Hybrid Poplars"

_plants, 2020, doi:10.3390/plants9040451_

Round 1

Reviewer 1 Report

In this study, the author reported that overexpression of Arabidopsis pPLAIIIα affected lignification in Arabidopsis and hybrid poplar. However, the author already reported that overexpression of PgpPLAIIIβ, which is 61.3% amino acid sequence identity with AtpPLAIIIα, affected ligninfication in Arabidopsis and hybrid poplar in Jang and Lee (2020). In addition, effect of overexpression of AtpPLAIIIβ, which is ~73% amino acid sequence identity with AtpPLAIIIα, in Arabidopsis had been reported in detail by Li et al. (2011). Therefore, the results indicated in this study were expected and not new. In this study, attention to ethylene production was a new viewpoint. However, the author just showed that expressions of ethylene synthesis-related genes changed in AtpPLAIIIα-OE plants but did not deepen it. I think ethylene production should be measured in AtpPLAIIIα-OE plants to know ethylene production is increased or decreased resultingly. If it is hard to measure ethylene production, expression of ethylene marker genes should be analyzed for further information about ethylene production. For instance, AtERF14, responsive to ethylene, and ERF1, responsive to ethylene and jasmonic acid, are candidates (Onate-Sanchez and Singh 2002 Plant Physiology 128: 1313-1322). Then relationship between change of ethylene and lignification should be considered.

As mentioned in the paperabove, AtpPLAIIIα and AtpPLAIIIβ have high similarity of amino acid sequence. The author should discuss the commonality and difference between AtpPLAIIIα-OE plants analyzed in this study and AtpPLAIIIβ-OE plants reported in previous paper (Li et al. 2011). Overall, the manuscript has several insufficient points.

  1. Others:

    1. The names of genes are obscure. I recommend to use combination of binominal name and gene name (e.g. AtpPLAIIIα, PgpPLAIIIβ) to all gene name. Gene name in the title of this paper is also obscure.

    1. In this paper, several pPLA genes were mentioned. The author should show the phylogeny tree of the mentioned genes (AtpPLAI, AtpPLAIIα, β, γ, δ, AtpPLAIIIα, β, γ, δ, OspPLAIIIα, PgpPLAIIIβ, PtpPLAIIIβ) for easy understanding.

    1. Abbreviation should be placed after full description. e.g. P2,L38, pPLAIII-OverExpression (OE). P6,L137, DNA polymerase change reaction (PCR).

    1. Figure 2. Illustration of the plant and the fonts in instructions are too small to read.

    1. About all AtpPLAs, I recommend to show AGI# (e.g. At2g39220).

    1. Table S1. PtpPLAIIIs were not shown their expressions by real time PCR method. However, Primers of PtpPLAIIIs were listed.

Author Response

it is attached as PDF file

Reviewer 2 Report

The manuscript by Jang and Lee reports that a patatin-related phospholipase pPLAIIIa may affect lignification of xylem in plants. They showed that, overexpression of Arabidopsis pPLAIIIa slightly reduces lignin content in Arabidopsis and poplars. They also showed a decrease in transcript level of several genes in lignin biosynthesis pathway and it is related to ethylene biosynthesis. The topic is of interest to the field of plant lipid metabolism and lignin biosynthesis. However, I would suggest the authors to improve the manuscript by clarifying the following major and minor concerns:

Major comments:

  1. Is activity of pPLAIIIa also induced in overexpression lines of pPLAIIIa? Lipid profiles in knockout mutant plants and overexpression lines should be analyzed.
  1. How does pPLAIIIa regulate genes in lignin and ethylene biosynthesis?
  1. Phenotypic analysis: What is phenotype of knockout mutant plants and overexpression lines of pPLAIIIa in Arabidopsis? 
  2. As seen in Figure 4, overexpression of Arabidopsis pPLAIIIa in poplar showed a dwarf phenotype, but lignin content of overexpression of pPLAIIIa in Arabidopsis and poplars did not show a big reduction compared to the WT as shown in Figure 2 and 5. Studying the phenotype from different approaches would be better to explain why overexpression of pPLAIIIa produces such the dwarf phenotype.
  1. In Figure 3, they checked expression level of genes related in ethylene and lignin biosynthesis in three overexpression lines: #6, #8 and #13 while they only used line # 8 and #13 in Figure 1 and Figure 2. They should include line #6 for analysis in Figure 1 and 2.

Minor comments:

  1. All figures would appear better if the author pay more attention on arrangement and formatting and use font style/font size consistently.
  2. Line #200: some words are not in a consistent font size, for example: “from Arabidopsis…”.
  3. Line #69-70, Figure 1B only showed qRT-PCR data of two overexpression lines, not three. Please correct it.

Author Response

It is attached as PDF file

Round 2

Reviewer 1 Report

In this revised paper, the insufficient points were revised mostly. However, it includes several minor points indicated below. I request their revisions.

P5,L110.

What is "Figure58"?

P16,L264.

C2H4 -> ethylene (unification of description).

P12,L198.

”Results" chapter did not include Figure 8. For instance, I propose that "2.6. Lignin Autofluorescence and YUCCA8 Gene Expression in AtpPLAIIIα-OE" is changed to "2.6. Lignin Autofluorescence and Expression of YUCCA8 and ERF Genes in AtpPLAIIIα-OE", and expression of ERF genes is described in this paragraph.

Author Response

In this revised paper, the insufficient points were revised mostly. However, it includes several minor points indicated below. I request their revisions.

P5,L110.

What is "Figure58"?

[Response] it is written as Figure 2B and I confirmed it again.

P16,L264.

C2H4 -> ethylene (unification of description).

[Response] As your comment, it is revised consistently. 

P12,L198.

”Results" chapter did not include Figure 8. For instance, I propose that "2.6. Lignin Autofluorescence and YUCCA8 Gene Expression in AtpPLAIIIα-OE" is changed to "2.6. Lignin Autofluorescence and Expression of YUCCA8 and ERF Genes in AtpPLAIIIα-OE", and expression of ERF genes is described in this paragraph.

[Response]As your comment, it is revised by adding the below sentence under your proposed title,

‘Overexpression of YUCCA8 and YUCCA9 led to strong lignification of aerial plant tissues, and it is linked to increased levels of ethylene production [28]. Constitutive ectopic expression of AtpPLAIIIα was resulted in down regulation of lignin biosynthesis-related gene expressions (Figure 3B), and modulated mRNA levels related to ethylene biosynthesis (Figure 3D). To go one step further, ethylene-responsive marker genes such as ERF1 and AtERF14 [29] in Arabidopsis as well as PtEFR1 and PtERF14 in poplar were quantified by qRT-PCR in different organs and were all confirmed to be upregulated by AtpPLAIIIα overexpression (Figure 8). It suggests that at least in stem of AtpPLAIIIα-OE, the lignification is related with the ethylene biosynthesis.’

Reviewer 2 Report

I think the authors have paid efforts to revise the previous version of manuscript and now it is improving significantly.

There are only a few minor comments:

  1. Please spell out all name of the genes used in the study if any is missing, especially genes in lignin biosynthesis and ethylene biosynthesis.
  2. [Optional] The author just can provide a simple scheme of lignin biosynthesis partway so that it is easy to follow.
  3. In Figure 3, use AtPAL1, At4Cl…so on for all genes in lignin and ethylene biosynthesis as in consistence with Figure 6.
  4. If the authors use AtpPLAIIIα, please use it consistently.

Author Response

I think the authors have paid efforts to revise the previous version of manuscript and now it is improving significantly.

There are only a few minor comments:

  1. Please spell out all name of the genes used in the study if any is missing, especially genes in lignin biosynthesis and ethylene biosynthesis.

[Response] As your comment, it is revised as below,

Line 164-169. phenylalanine ammonia-lyase 1 (PAL1), cinnamate-4-Hydroxylase 1 (C4H1), 4-coumarate-CoA ligase (4CL), Hydroxycinnamoyltransferase 1 (HCT1), 4-coumarate 3-hydroxylase 3 (C3H3), caffeoyl-CoA O-methyltransferase 1 (CCOAOMT1), cinnamoyl-CoA reductases 2 (CCR2), catechol-O-methyltransferase 2 (COMT2), and cinnamyl alcohol dehydrogenase 1 (CAD1)

Figure 3. Expression levels of lignin and ethylene biosynthetic genes by qRT-PCR.

(A) Lignin biosynthesis pathway, and (B) relative mRNA levels of genes related to lignin biosynthesis; PAL: phenylalanine ammonia-lyase 1; C4H: cinnamate-4-hydroxylase; 4CL: 4-coumarate-CoA ligase; HCT: hydroxycinnamoyltransferase; C3H: 4-coumarate 3-hydroxylase; CCoAOMT: caffeoyl-CoA O-methyltransferase; CCR: cinnamoyl-CoA reductases; COMT: catechol-O-methyltransferase; F5H: coniferaldehyde 5-hydroxylase; CAD: cinnamyl alcohol dehydrogenase. (C) Simple schematic diagram of ethylene biosynthesis pathway and (D) mRNA levels of genes related to ethylene biosynthesis; SAM: S′adenosyl methionine, ACC: 1-aminocyclopropane-1-carboxylic acid. Data represent the mean ± SE of three independent replicates P < 0.05(*) and P < 0.01(**), respectively.

  1. [Optional] The author just can provide a simple scheme of lignin biosynthesis partway so that it is easy to follow.

[Response] As your comment, it is revised by adding Figure 3A.

  1. In Figure 3, use AtPAL1, At4Cl…so on for all genes in lignin and ethylene biosynthesis as in consistence with Figure 6.

[Response] As your comment, it is revised. In main text, the gene names were also all revised.

  1. If the authors use AtpPLAIIIα, please use it consistently.

[Response] As your comment, it is revised throughout the main text in MS.